# A Mini-Review of Recent Developments in Plenoptic Background-Oriented Schlieren Technology for Flow Dynamics Measurement

**Yulan Liu, Feng Xing, Liwei Su, Huijun Tan * and Depeng Wang ***

College of Energy and Power Engineering, Nanjing University of Aeronautics and Astronautics, Nanjing 210016, China
* Correspondence: tanhuijun@nuaa.edu.cn (H.T.); depeng.wang@nuaa.edu.cn (D.W.)

**Abstract:** To uncover the underlying fluid mechanisms, it is crucial to explore imaging techniques for high-resolution and large-scale three-dimensional (3D) measurements of the flow field. Plenoptic background-oriented schlieren (Plenoptic BOS), an emerging volumetric method in recent years, has demonstrated being able to resolve volumetric flow dynamics with a single plenoptic camera. The focus-stack-based plenoptic BOS system can qualitatively infer the position of the density gradient in 3D space based on the relative sharpness of the refocused BOS image. Plenoptic BOS systems based on tomography or specular enhancement techniques are realized for use in high-fidelity 3D flow measurements due to the increased number of acquisition views. Here, we first review the fundamentals of plenoptic BOS, and then discuss the system configuration and typical application of single-view and multi-view plenoptic BOS. We also discuss the related challenges and outlook on the potential development of plenoptic BOS in the future.

**Keywords:** plenoptic background-oriented schlieren; volumetric imaging; flow measurement





## 1. Introduction

By exploiting the relation between refractive index and density, schlieren-based flow visualization techniques can sensitively reveal inhomogeneous density variations of the measurement region in a qualitative and non-invasive manner [1]. One of them, BOS, requires only a background pattern as a reference, and then determines the change in density by calculating the displacement of the two images captured with and without density interference [2]. BOS has significant advantages such as a simple hardware construction, convenient calibration, and the free limitation in the measurement window by the size of optical components, thereby visualizing the variable-density flow and measuring the related refractive index field and density field quantitatively. So far, BOS has successfully demonstrated its application in revealing the field of supersonic flow [3], combustion [4], plasma [5], and other complex flows.

Since most flows are inherently 3D, this has driven the development of BOS techniques toward three-dimensional, quantitative, and high-spatial- and -temporal-resolution measurement. Tomographic BOS is one of such 3D systems with a large measurement range and high measurement accuracy. Based on the number of cameras employed in the system, tomographic BOS can be categorized into multi-camera configurations [6,7] and single-camera configurations along multiple views [8]. In past studies, single-camera configurations using up to 36 different views resulted in a significant loss of temporal resolution, as well as additional costs for precisely moving the camera or rotating the model. In contrast, multi-camera (8+) configurations make the whole system bulky and impractical for applications with limited optical access.

The recent advance of plenoptic BOS [9,10] has greatly simplified 3D-BOS systems, enabling single-view volume reconstruction and the measurement of 3D density fields

with only one plenoptic camera. This plenoptic camera captures both spatial and angular information using a microlens array (MLA), which provides the ability to change the perspective angle or focal plane position in an integrated manner during the post-processing stage. In comparison to traditional BOS, plenoptic BOS can produce a greater depth of field (DOF) while retaining a large numerical aperture for efficient light collection [11]. So far, plenoptic BOS has been successfully applied to the 3D imaging of flame plumes [12], shock wave turbulent boundary layer interactions [13], and supersonic free air jets [14].

In this review, we introduce the fundamentals of plenoptic BOS and emphasize its imaging ability to change perspective and refocus. Then, we summarize plenoptic BOS applications of single-view systems based on focus stacks, and multi-view systems based on tomography or mirror enhancement technology. Finally, we also discuss the related challenges and outlook on the potential of plenoptic BOS to bring about further developments.

## 2. Fundamentals

### 2.1. Background-Oriented Schlieren

BOS is a technology independently proposed by Dalziel [15], Richard [16], and Meier [17] around 2000 that only requires an imaging system, background pattern, light source, and a computer for image processing. The core idea is to use the displacement of the background pattern to describe the deflection of light. A typical BOS experimental arrangement is shown in Figure 1a. When there is no density perturbation, light from a point on the background is focused through the main lens at a location on the image sensor (shown as blue), and the blue line represents the center of the light cone entering the aperture of the main lens. The image captured at this point is usually referred to as the reference image. When an inhomogeneous density field exists, refraction occurs, causing light from the same point on the background with a different angular range to enter the main lens (shown as red). This is manifested as a displacement $\Delta$ of the background pattern imaging position on the reference image. According to the geometry of the experimental configuration, the relationship between the deflection angle $\varepsilon$ and the displacement of the background pattern is obtained (1):

$$\varepsilon = (\frac{Z_a}{Z_d} + 1)(\frac{\Delta_{y\prime}}{f_m})\tag{1}$$

where $Z_a$ is the distance of the schlieren object from the lens, $Z_i$ is the distance between the lens and the imaging surface, $Z_d$ is the distance between the center of the schlieren object and the background plate, $\Delta_{y\prime}$ is the magnitude of vector displacement in the imaging sensor plane, and $f_m$ is the focal length of the main lens. At a larger $Z_a/Z_d$ (i.e., when the plane of the schlieren object is close to the background plane), the sensitivity or magnitude of the measured deflection decreases. The most frequently utilized technique to estimate displacements is cross-correlation, a widely used algorithm in particle image velocimetry (PIV) for particle displacement computation [18]. In addition, optical flow [19], dot tracking [20], and modulation–demodulation type [21] algorithms can also be applied to BOS. The local magnitude and direction of the picture displacement can be detected by these algorithms. The density gradient's size and location in relation to the camera and background determine the displacement's magnitude, which is an integrated line-of-sight quantity. The interpretation of such displacements collected with a single camera is usually restricted to qualitative analysis, leaving the 3D aspects of the flow unclear, because one cannot predict in advance the range of depths over which refraction occurs. The refractive index $n$ distribution is obtained by substituting the displacement into the Poisson Equation (2) solution, and then the Gladstone–Dale Equation (3) is used to obtain the density distribution [22].

$$\frac{\partial^2 n}{\partial x^2} + \frac{\partial^2 n}{\partial y^2} = K[\frac{\partial \Delta_x}{\partial x} + \frac{\partial \Delta_y}{\partial y}]\tag{2}$$

$$n - 1 = G\rho\tag{3}$$

where $\Delta_x$ and $\Delta_y$ are the background pattern displacements in different directions, $K$ is a constant related to the experimental configuration, $G$ is the Gladstone–Dale constant, and $\rho$ is the density.

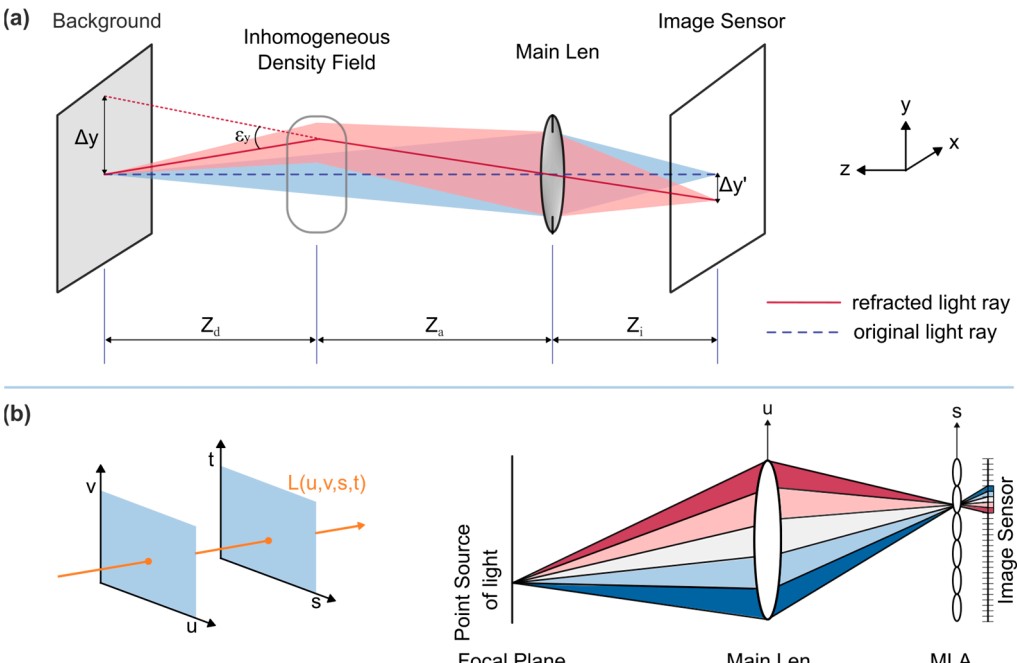

**Figure 1.** Principle diagram of plenoptic BOS. (**a**) Geometric relationship in the BOS configuration. (**b**) 4D light-field model (**left**), and schematic of imaging with a plenoptic camera (**right**).

### 2.2. Light-Field Imaging and the Plenoptic Camera

The light field was first formally defined by Gershun [23] in 1936 as a collection of light rays in space. Later, Levoy and Hanrahan [24] developed a four-dimensional (4D) parametric light-field model, as shown in Figure 1b (left) and represented by Equation (4), in terms of rays intersecting two planes: $(u, v)$ and $(s, t)$. The light ray itself is defined by its position across two planes. The irradiance of a light ray, represented by the four co-ordinates $u$, $v$, $s$, and $t$, is represented by the letter $L$.

$$L = L(u, v, s, t) \tag{4}$$

The plenoptic camera samples a 4D light field by adding the MLA at the focus of the main lens, allowing 3D imaging using only one device. In 1992, Adelson and Wang [25] developed a plenoptic camera, which was further improved by Ng et al. [26] in 2005. A traditional camera directs light from a point on the focal plane to a corresponding single-point pixel on the image sensor. In contrast, a plenoptic camera's MLA directs light from a specific point on the focal plane to different pixels on the image sensor, depending on the angle at which the light enters the main lens. This is represented by the five different-colored subsets of light entering the main lens in Figure 1b (right). Pixels encode angular light-field data using $(u, v)$ co-ordinates, while microlens plane positions encode spatial information using $(s, t)$ co-ordinates. A plenoptic image captures spatial and angular data, allowing images to be generated from different viewing angles and focal planes during post-processing. By collecting light rays in each microlens that pass through the same position $(u, v)$ in the plane of the main lens, we can change the perspective. By projecting the light field onto a new image plane for integration, we can refocus the image. It is significant to note that volumetric calibration of the plenoptic camera is essential for high-quality flow field reconstruction. Hall et al. [27] proposed two polynomial-mapping-function-based volumetric calibration methods, including volumetric dewarping and direct light-field calibration, which can be used with a minimal understanding of lens parameters.

*2.3. Plenoptic Background-Oriented Schlieren*

Klemkowsky et al. [10] integrated BOS technology with the distinctive imaging capabilities of plenoptic cameras, thereby introducing plenoptic BOS as a new addition to the schlieren technology family. Plenoptic BOS, in contrast to typical single-camera BOS, has the ability to sample multiple lines-of-sight at the same time. The displacements observed from each line-of-sight are combined to construct a four-dimensional displacement field. This field is a vector function organized in a manner similar to the original light field collected in a raw plenoptic image. Focused BOS images are rendered via displacement fields, and the images are characterized as narrow DOF slices of density gradient fields. Plenoptic BOS, unlike focused schlieren methods, does not require a human adjustment of the focal plane during data collection. Instead, it comprehensively alters the position of the focal plane during post-processing, allowing the acquisition of all focal planes in a single photograph.

The DOF refers to the finite distance in front of and behind the nominal focal plane where objects appear to be in focus in an imaging system. As the distance from the focal point grows, objects progressively lose their sharpness and become increasingly blurred. Disregarding diffraction, the DOF can be estimated theoretically by determining the limits of depth that are close and far away, based on geometric optics. According to Kingslak [28], Equations (5) and (6) demonstrate that these limits are established based on unchanging camera characteristics. Here, $d_a$ represents the diameter of the main lens aperture, $l_o$ represents the distance to the object plane, and $c_o$ represents the circle of confusion in object space. The DOF can be calculated by finding the difference between the values of $z_{near}$ and $z_{far}$, as represented in Equation (7):

$$z_{near} = \frac{d_a l_o}{d_a + c_o} \tag{5}$$

$$z_{far} = \frac{d_a l_o}{d_a - c_o} \tag{6}$$

$$DOF = z_{far} - z_{near} \tag{7}$$

The sensitivity of a BOS system is largely contingent upon the optical arrangement of the experiment, which determines the system's ability to detect minor density gradients. Bichal et al. [29] found that, as the magnification and circle of confusion increased, both the system sensitivity and the DOF increased similarly. In order to optimize the sensitivity, an experimental setup positions the backdrop at the furthest plane within the DOF and the schlieren object at the nearest plane within the DOF [9]. This ensures that both the background and the schlieren object are in sharp focus. The DOF can be computed by the effective aperture diameter $d_{eff}$, using Equation (8). In this equation, $p_p$ represents the pixel pitch, $l_i$ represents the distance in image space, and $f_\mu$ represents the focal length of a microlens. The term $d_a$ used in Equations (5)–(7) for calculating the DOF is replaced by the effective diameter.

$$d_{eff} = \frac{p_p l_i}{f_\mu} \tag{8}$$

## 3. Plenoptic BOS System

According to the number of imaging views that the plenoptic cameras (real or virtual) are arranged relative to the schlieren object, the developed plenoptic BOS systems can be categorized into two categories: single-view and multi-view systems.

*3.1. Single-View Plenoptic BOS Imaging*

3.1.1. Imaging Process

Early single-view plenoptic BOS systems used focus stacks for qualitative 3D density gradient localization [30]. The relevant workflow is shown in Figure 2. Plenoptic images are first taken in the presence and absence of density perturbations, respectively. Sub-

sequently, these two images were decoded to generate the same number of perspective views. The cross-correlation of the two sets of perspective pictures yields a 2D displacement field of the background in each perspective. Finally, the BOS focus stack is created by a refocusing program that sums the displacement information by equating it to intensity.

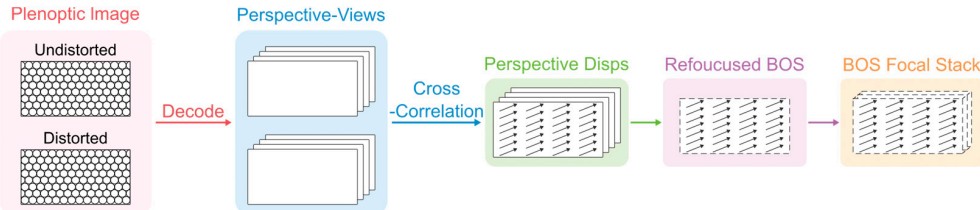

**Figure 2.** Flow chart of the single-view plenoptic BOS process.

Klemkowsky et al. [11] conducted a direct comparison between conventional and plenoptic BOS systems. Upon comparing the two, it was noted that plenoptic BOS offers several advantages over conventional BOS. These advantages include an increased DOF, improved light collection efficiency, a simplified focus operating mode, and the ability to determine the depth location of a density object using just one camera. However, the trade-off results in a signal-to-noise ratio that is four to five times poorer compared to traditional BOS. Additionally, in the conventional system, the number of available pixels is determined by the pixels on the camera sensor, which is $6600 \times 4400$; in the plenoptic system, the number of available pixels is determined by the quotient of the camera's pixel number and the number of pixels behind each microlens. When the number of pixels behind each microlens is $14 \times 14$, the resultant available pixels are $471 \times 362$, leading to a 14-times decrease in resolution. Regarding the temporal resolution, it highly depends on the frame rate of the imaging camera used in the BOS system, which may vary depending on the setup.

### 3.1.2. Applications

The single-view plenoptic BOS system has been used for the imaging of a laminar heated jet [10], shock-turbulent boundary layer interaction [13], and an underexpanded jet [31]. Low-resolution 3D positioning can be achieved based on image sharpness in the focus stack, and Figure 3 shows examples of applications for such stack.

Klemkowsky [10] conducted a demonstration experiment in the advanced flow diagnostics laboratory (AFDL) to qualitatively investigate the density gradient generated by two heat sources located at different depths concurrently. A pair of flames was employed to create an ascending plume of heated air that interacted with the ambient air in the space. A 5 mm-diameter portable lighter was positioned 365 millimeters in front of the nominal focal plane. A Bunsen burner fueled by natural gas with an inner diameter of 11 mm was positioned at a distance of 177 mm beyond the nominal focal plane. The two flames were positioned at different locations within the field of view, as depicted in a center perspective image in Figure 3a (left). A Nikon AF-135 mm primary lens was used to set up an IMPERX Bobcat 6620 (29 MP) camera (IMPERX, Boca Raton, FL, USA) equipped with a KAI-29050 CCD image sensor (TrueSense Marketing, Warrendale, PA, USA). AFDL modified the camera with a $471 \times 362$ hexagonal MLA, where each microlens has a spacing of 77 μm and a focal length of 308 μm. The raw plenoptic image had a magnification of approximately $-0.11$ and an f-number of f/4. The raw images are processed using an internally developed open-source MATLAB package called the Light-Field Imaging Toolkit (LFIT) [32]. This code offers functionalities for calibrating, rendering perspective views, and generating focal stacks.

Figure 3a (middle) and Figure 3a (right) display two focused BOS pictures rendered at two synthetic focal planes that correspond to the estimated depth position of each flame. In Figure 3a (middle), the portable lighter in the foreground provides a clear picture of the detailed interaction between the hot air plume and the room-temperature air. The intricate

information is obscured when the synthetic focal plane is adjusted to the emission plume associated with the Bunsen burner in Figure 3a (right). Clearly, the level of detail in each flame structure is visible when it is in focus, but becomes hazy when it is out of focus. These qualitative findings allow us to deduce depth from focused BOS images.

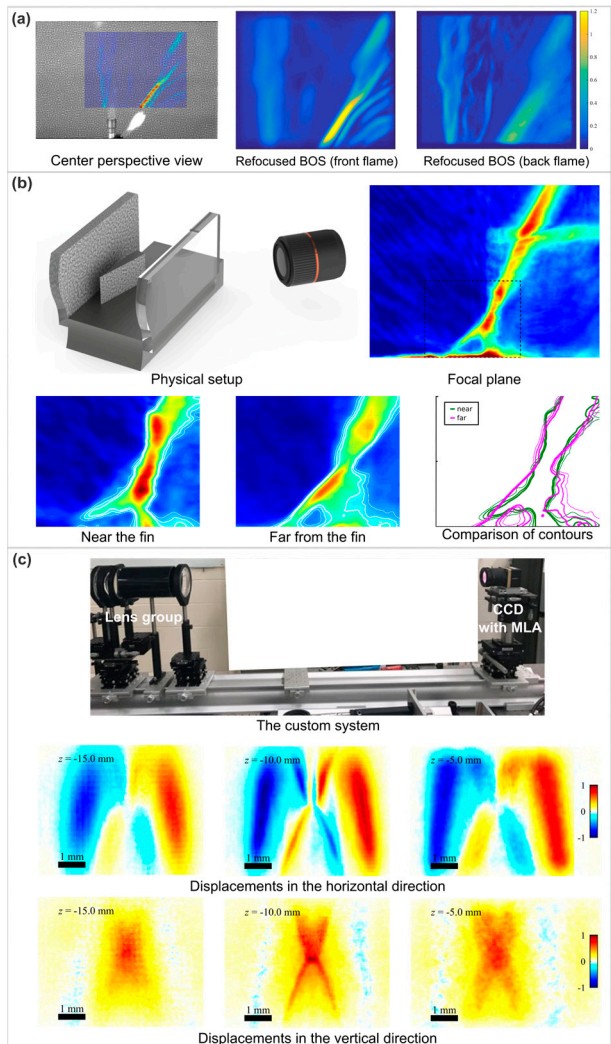

**Figure 3.** Single-view plenoptic BOS applications. (**a**) The displacement of the center perspective (**left**), and the refocused BOS images at the front flame (**center**) and the back flame (**right**). Colorbar and axes are in pixels. Reprinted with permission from reference [10]. (**b**) The physical arrangement in the SBLI experiment and the shock foot contours obtained at different focal planes and their comparison. Reprinted with permission from reference [13]. (**c**) The custom system (**top**) and the refocused BOS displacements in the horizontal (**center**) and vertical (**bottom**) directions. Reprinted with permission from reference [14].

Shock/boundary layer interaction (SBLI) is an inherently 3D unsteady phenomenon [33,34]. Previous research has primarily focused on studying SBLI using either line-of-sight approaches or two-dimensional methods. Both of these techniques are fundamentally limited to quasi-2D representations; hence, they cannot provide a comprehensive range of information about the flow field. Clifford et al. [13] studied multiple focal planes of a swept SBLI using a plenoptic BOS technique designed to visualize density gradients in the SBLI. The experiments were conducted in the supersonic wind tunnel facility located at the Florida Centre for Advanced Aero-Propulsion of Florida State University (FCAAP-FSU). The dimensions of the test section were 76.2 mm in height, 101.6 mm in width, and 393.7 mm in length. The experiments were conducted at Mach 2, with a stagnation temperature of 296 K and a stagnation pressure of

350 kPa. The dimensions of the fin model were 76.2 mm in length and 37.5 mm in height. The angle of attack was adjusted to 15°, resulting in the formation of an oblique shock wave at an angle of 45° from the fin apex (Figure 3b (top left)). The plenoptic camera was situated at an angle of roughly 45° from the fin apex, aligning the optical axis with the shock propagation. The background generated using the wavelet noise algorithm [35] was printed on smooth adhesive-backed paper. Instead of positioning the background behind the scene, it was affixed to the model geometry inside the supersonic tunnel. The camera was the same as in the experiment above.

The 3D feature of the swept interaction is evident in Figure 3b (bottom), which corresponds to the dashed region seen in Figure 3b (top right). In Figure 3b (bottom), the colormap representing the density gradient magnitude is overlaid with a small number of selected contours, which have been isolated for comparison. The shock foot's size and shape change in relation to the distance from the fin. In close proximity to the fin, the $\lambda$ structure exhibits a relatively concise length and exhibits rounded characteristics. As one moves further from the fin, the height of the structure grows and its characteristics become straighter and more triangular in shape. These findings reveal the interacting shock foot and its spatial evolution, and are verified by plenoptic particle image velocimetry (PIV) [36,37].

An experiment was conducted at Sandia National Laboratories to capture 3D images of shock structures in an underexpanded free air jet generated by a convergent–divergent nozzle [14]. The system functioned with a back pressure of 700 psig, leading to an exit Mach number of around 3.3. The jet's exit diameter was 6.35 mm, while the ratio of the exit area to the throat area was roughly 8.26. The DOF measurements for the perspective and focused BOS images were 3.9 mm and 1.3 mm, respectively. Given the significant magnification in this configuration, the background pattern needed to be transmitted to the DOF of the plenoptic system by employing a custom lens system (Figure 3c (top)) situated in free space.

The center row of Figure 3c displays the horizontal displacement of refocusing, whereas the bottom row displays the vertical displacement. The color corresponds to the magnitude of displacement. The air jet was positioned at a distance of $-10$ mm from the nominal focal plane and the measured displacement is maximum at this plane. Figure 3c also demonstrates that the observed displacements become blurred when $z = \pm 5$ mm. The experimental results indicate a spatial resolution of approximately 25 $\mu$m, along with the ability to perform 3D imaging. Therefore, it is possible to accomplish qualitative 3D localization based on picture sharpness within the focus stack.

### 3.2. Multi-View Plenoptic BOS Imaging

Historical studies have shown that a single plenoptic camera in conventional plenoptic BOS systems is not sufficient for accurate BOS reconstruction [38]. Therefore, recent studies have explored plenoptic BOS systems based on multi-camera tomography technology and single-camera mirror-enhanced dual-view to achieve higher-accuracy 3D density field reconstruction, respectively.

### 3.2.1. Tomographic Plenoptic BOS

Davis [38] has developed a tomographic BOS system using 1–4 plenoptic cameras and an iterative simultaneous algebraic reconstruction technique (SART) [39] algorithm to achieve the quantitative volume reconstruction of known 'static flow' fields. The workflow starts with the calibration of the microlens [40] and volume [27]. Next, the BOS measurement dataset is constructed through the Farneback optical flow algorithm [41] provided by OpenCV. Then, the volume and masking settings are completed. Finally, the calibration information, displacement information, and setting information are used together for a reconstruction implementation based on the SART algorithm. Previous research indicates that achieving a high-quality outcome requires both a wide range of angles and a uniform distribution of projections. This study examined the implications of utilizing plenoptic

cameras, which encompassed two distinct investigations. The impact on the reconstructed volume is detected by initially altering the number of perspective views employed per camera, followed by modifying the total number of plenoptic cameras used.

Figure 4a shows a top view schematic of the experimental setup. Transparent cylinders made of dimethylpolysiloxane are utilized as stationary objects submerged in a field with a refractive index that matches their own, and work as substitutes for basic characteristics that can be noticed in a flow field. The two 6.35 mm-radius cylinders are 30 mm apart and the octagonal tank was constructed from acrylic plates. The interior tank width, measured as the space between the two opposing walls, was 184 mm. Each wall had a length of 76.2 mm on the inside and a height of 254 mm. The backdrop pattern was oriented at a right angle to the optical axis of each camera on the opposite wall. A total of four cameras were used in the experiment: two IMPERX B6620 cameras and two IMPERX B6640 cameras. Each camera is equipped with a KAI-29 050 29 MP CCD sensor with a bit depth of 12 bits. The camera contains a hexagonally packed MLA with dimensions of $471 \times 362$. Each microlens has a focal length of 0.308 mm and a pitch of 0.077 mm. The optical axis of each of the four cameras was spaced apart at an angle of $45°$. The function generator was used to trigger all cameras simultaneously. All four plenoptic cameras were equipped with a primary lens with a focal length of 60 mm. These cameras had an approximate nominal magnification of $-0.3$ and were set to exposure at 40 ms.

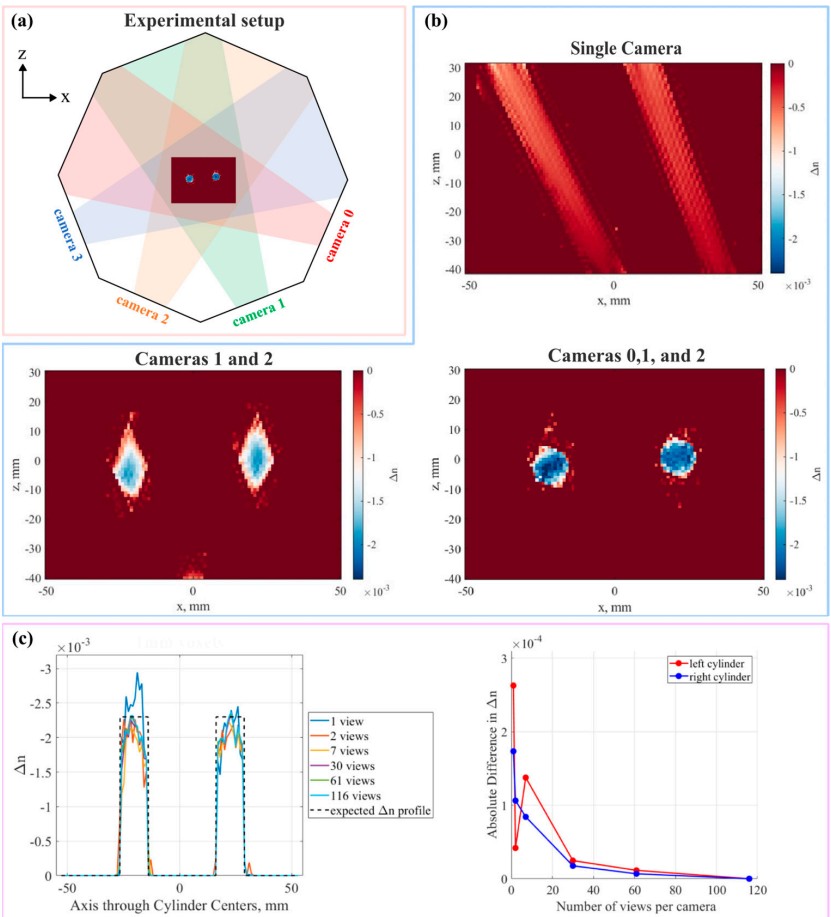

**Figure 4.** Tomographic plenoptic BOS. Reprinted with permission from reference [38]. (**a**) Reconstruction of four cameras overlaid on the schematic of the experimental setup for reference. (**b**) Reconstructions for the different camera configurations. (**c**) Reconstruction varying the number of perspective views used by each camera. Profile along cylinder centers (**left**) and $|\Delta n_{116} - \Delta n|$ (**right**).

Figure 4b shows the reconstructed x–z center plane slices for a different number of cameras. Obviously, the restricted angular data captured by a single camera lead to the

significant elongation of the rebuilt object, which greatly compromises the quality of the reconstruction. Irrespective of the angular span between the two cameras, there are insufficient data to remove the artifacts from the solution. The addition of a third camera significantly enhanced the accuracy of the rebuilt objects, bringing them closer to the true representation. Moreover, the reconstruction outcomes with varying numbers of perspective views per camera are contrasted in a four-camera setup (Figure 4c). Reconstruction artifacts were reduced and the solution was significantly improved by just increasing the number of views from one to two. And the solution gets smoother as the number of views increases. For the 30- and 61-view examples, the differences are less than $2 \times 10^{-5}$, which is a significant reduction from the disparities calculated with fewer views per camera (Figure 4c (right)). It is shown that the reconstruction quality is dependent on the number of perspective views used per plenoptic camera and the total number of cameras used in the reconstruction process. These observations show that the usage of plenoptic cameras may perform better than an equivalent number of conventional cameras if the number of cameras available is limited (for example, because of optical access limits).

### 3.2.2. Mirror-Enhanced Isotropic-Resolution Dual-View Plenoptic BOS

Increasing the number of plenoptic cameras to create a tomographic setup and subsequently capturing the density field from various perspectives can enhance the axial resolution. However, this approach unavoidably incurs supplementary expenses in terms of budget, system alignment, and calibration, as well as additional constraints on system allocation due to limited space and optical access. The recent study developed a dual-view isotropic-resolution plenoptic BOS (ISO plenoptic BOS) system [42] that utilized a mirror to generate a second image view of the region of interest, thereby effectively enhancing the axial resolution of conventional plenoptic BOS systems and achieving isotropic spatial resolution using just one camera. The system comprises one mirror, two background plates, and a single plenoptic camera (Figure 5a). Due to the 45° angle between the mirror and the camera's sensor plane, the camera is able to capture the density field in the lateral direction by imaging it through the mirror, which is equivalent to adding a lateral virtual view to the field of view (FOV). The left and right halves of a camera's sensor array are used to acquire front view (view 1 in Figure 5a) and side view (view 2 in Figure 5a) images, respectively. By fusing the data from view 2 with view 1, the axial resolution of view 1 can be enhanced, leading to an isotropic spatial resolution [43]. This is because light-field imaging has a superior lateral resolution compared to its axial resolution. In the BOS experimental setup, both the patterns and schlieren object are within the depth of field of the plenoptic camera, which requires that the optical path difference of the dual-view version be as small as possible. Hence, the pattern in view 2 needs to be positioned near the mirror, which would inevitably reduce the FOV.

The system employed a Lytro Illum light-field camera equipped with a resolution of $7728 \times 5368$ pixels, an MLA of $625 \times 434$, and a fixed aperture of F/2.0. In the validation experiment of dual candle flames, the two candle flames were positioned in predetermined locations so that one flame was situated behind the other when observed from the front (Figure 5b). The system's nominal focal plane was positioned at background pattern 1, the front candle (A) was positioned 140 mm in front of the nominal focal plane, and the two candles were positioned 70 mm apart and 130 mm distant from background pattern 2. The imaging area is $100 \times 100 \times 100$ mm$^3$.

The candle flame density field reconstructed using single-view imaging has a clear elongation along the axial direction. Therefore, it is not possible to differentiate between the two flames (flames A and B) that are axially aligned in the front-view image (Figure 5c (top left)). Conversely, by transforming the front view's axial resolution into the lateral resolution of the lateral view, the lateral view can differentiate between the two flames (Figure 5c (top right)). Figure 5b (bottom) shows the flame profile from different views. It is evident that the reconstructed outcome from the dual-view system accurately detected two distinct peaks corresponding to the two flames. However, the profile obtained from

the conventional method failed to distinguish two peaks. In addition, this study also reconstructed the high-resolution Mach disk density field in the $8 \times 8 \times 14 \ \text{mm}^3$ area, which verified the feasibility of the system for supersonic flow imaging. All results demonstrated that the ISO plenoptic BOS system possesses decent scalability and is well-suited for high-resolution, large-field aerodynamic studies.

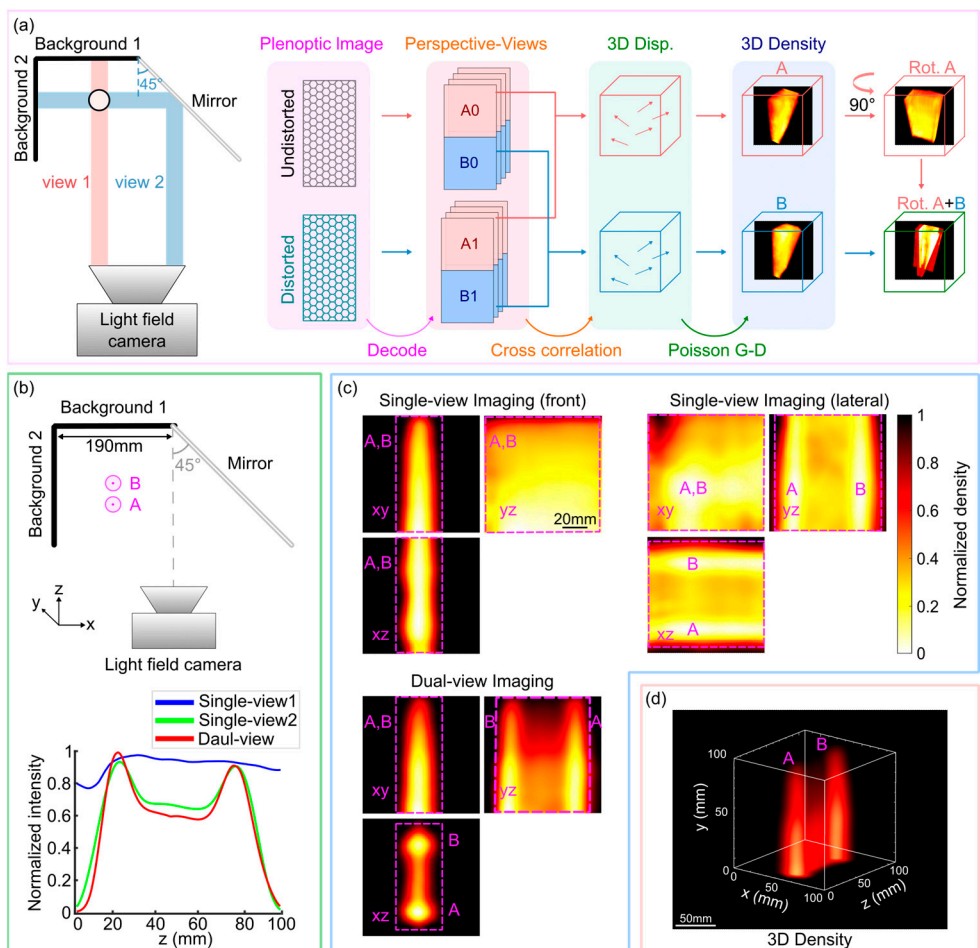

**Figure 5.** ISO plenoptic BOS. Reprinted with permission from reference [42]. (**a**) Experimental setup diagram and work flow of ISO plenoptic BOS. (**b**) Schematic diagram of the validation experiment with two candles and the flame profile in all views. (**c**) Comparison of densities reconstructed with front view, lateral view, and dual-view images (minimum intensity projection). (**d**) Reconstructed 3D density field.

## 4. Conclusions

By combining BOS and light-field imaging technology, plenoptic BOS provides a systematic and simple way to reconstruct 3D density fields. Compared with traditional 3D measurement systems, plenoptic BOS can simultaneously sample multiple lines of sight, which can reduce the cost of cameras, hardware synchronization, scanning, and data calculations, and is more advantageous in situations where optical access is limited.

The challenge of finite parallax baselines and spatial–angular resolution tradeoffs is the major issue for plenoptic systems. In addition, the improvement of spatial resolution and the capture of small and transient disturbances are also challenges faced by plenoptic BOS. Therefore, the plenoptic BOS system can be further optimized in the future in four aspects: MLA, BOS experimental setup, imaging speed, and data processing efficiency. MLA determines the spatial resolution of the rendered perspective view in the plenoptic BOS system. Developments in MLA fabrication have enabled cheaper, more precise, and more creative devices, such as nanoimprint MLA [44] and heterogeneous focal length

MLA [40], which contribute to the light-field spatial resolution. A reasonable adjustment of BOS parameters (e.g., sensitivity S) in the experimental setup helps to maximize the signal from a given schlieren object [45]. Data-processing methods can also help to improve the performance of plenoptic BOS. For instance, a more accurate displacement can be obtained by applying optical flow algorithm [19], which is more sensitive to sub-pixel displacement; a higher-resolution image can be reconstructed using a deconvolution algorithm [46], which can filter out the out-of-focus information on the refocused plane. The imaging speed is limited by the frame rate of the camera. An ultrafast camera can be used to increase the imaging frame rate for transient flow imaging [47]. Optically multiplexed schlieren videography capture time-resolved sequences in a single photograph by illuminating the sample with a unique rapid burst of spatially modulated light pulses, thus, in principle, enabling infinite frame rates [48]. In addition, deep learning can be applied to improve the speed and quality of light-field reconstruction [49] and displacement field reconstruction [50,51], making it possible to achieve instantaneous high-resolution volumetric imaging.

In the future, the possibility of the plenoptic BOS system for instantaneous high-resolution volumetric imaging can be explored, and the system can be further extended to flow field measurements at optically constrained sites containing complex 3D density fields, such as high-speed wind tunnels, for example, to reveal the underlying flow mechanisms.

**Author Contributions:** Conceptualization, Y.L. and D.W.; methodology, Y.L. and D.W.; resources, Y.L. and D.W.; writing—original draft preparation, Y.L. and D.W.; writing—review and editing, Y.L., D.W., H.T., F.X. and L.S.; visualization, Y.L. and D.W.; supervision, H.T. and D.W.; project administration, D.W.; funding acquisition, H.T. and D.W. All authors have read and agreed to the published version of the manuscript.

**Funding:** This work was funded by the Young Elite Scientist Sponsorship Program by CAST (YESS20210238), the National Natural Science Foundation of China (Nos. U20A2070, and 12025202), the startup of Nanjing University of Aeronautics and Astronautics (90YQR23004), the Fundamental Research Funds for the Central Universities (NO. NS2023007), and the Natural Science Foundation of Jiangsu Province (BK20230876).

**Data Availability Statement:** Not applicable.

**Conflicts of Interest:** The authors declare no conflicts of interest.

## Nomenclature

The following abbreviations and symbols are used in this manuscript:

| | |
|---|---|
| BOS | Background-oriented schlieren |
| DOF | Depth of field |
| MLA | Microlens array |
| PIV | Particle image velocimetry |
| LFIT | Light-field imaging toolkit |
| SBLI | Shock/boundary layer interaction |
| SART | Simultaneous algebraic reconstruction technique |
| FOV | Field of view |
| ISO | Isotropic |
| 2D | Two-dimensional |
| 3D | Three-dimensional |
| 4D | Four-dimensional |
| $\varepsilon$ | Angular ray deflection |
| $Z_a$ | Distance from schlieren object plane to the main lens |
| $Z_d$ | Distance from schlieren object plane to background plane |
| $\Delta_{y'}$ | Magnitude of vector displacement in the imaging sensor plane |
| $f_m$ | Focal length of main lens |
| $f_\mu$ | Focal length of the microlenses |
| $n$ | Refractive index |
| $\rho$ | Density |

| $K$ | Constant related to the experimental configuration |
| $G$ | Gladstone–Dale constant |
| $d_a$ | Diameter of the main lens aperture |
| $l_o$ | Object distance to nominal focal plane |
| $l_i$ | Image distance to nominal film plane |
| $c_o$ | Circle of confusion in object space |
| $z_{near}$ | Near depth of field limit |
| $z_{far}$ | Far depth of field limit |
| $d_{eff}$ | Effective diameter aperture used to render a perspective view |
| $p_p$ | Pixel pitch |
| psig | Pound per square inch |

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
