# Peer review of "A Mini-Review of Recent Developments in Plenoptic Background-Oriented Schlieren Technology for Flow Dynamics Measurement"

_aerospace, doi:10.3390/aerospace11040303_

Round 1

Reviewer 1 Report

Comments and Suggestions for Authors

The original intention was to advise the authors to change the title of the article and omit the word ‘mini-review’ from the title but this article is really a mini-review with all consequences. The fundamentals of plenoptic BOS method are presented in a shortly manner and the applications are shown up with a few examples. These shortcomings are partly compensated by the references, which gives close to comprehensive representation of the topic. This review will be a good starting point to dive into the related topic of plenoptic BOS technology. The manuscript is well written and can be accepted in the present form.

To be accurate it should be pointed on a few discrepancies:

1. Refs. [27] “Kingslake, R., Depth of Field. 1992.” Does this link refer to the chapter in the book Optics in Photography by Rudolf Kingslake?

2. See the line 208. The reference have to be inserted: “Clifford et al. [13] studied multiple focal planes of a swept SBLI”

Reviewer 2 Report

Comments and Suggestions for Authors

This work presents a review of the recent developments in plenoptic background oriented schlieren for flow field measurement. The review starts with the introduction of BOS and plenoptic imaging principles, followed by plenoptic BOS systems and applications. Several examples of experimental setups and the results were presented, and the remaining challenges and limitations were summarized in the conclusions. Overall, the manuscript was well-written and organized. The concepts were clearly explained, and the examples were well described with sufficient details. However, I'd like to provide a few suggestions to further improve this manuscript. 

1. A list of abbreviations and symbols should be included.

2. When mentioning about the limitations of plenoptic BOS, such as lower spatial and temporal resolution and signal-to-noise ratio as compared to single-camera conventional BOS, it is suggested to provide some range of the values, e.g., the temporal resolution is xx fps for plenoptic BOS, while it can achieve xx fps for conventional BOS.

3. Since the plenoptic BOS provide richer information, i.e., not only the spatial information but also angular information, it is more challenging to reconstruct plenoptic BOS, e.g., the increased computational complexity, the integration of multiple source of data, etc. This review should elaborate more on this aspect about reconstruction methods and challenges. 

4. There is a typo on figure 1. "Inbomogeneous Densuty Field".
